# Comparing pharmaceutical company payments in the four UK countries: a cross-sectional and social network analysis

Emily Rickard [ORCID], Emma Carmel, Piotr Ozieranski [ORCID]

Department of Social and Policy Sciences, University of Bath, Bath, UK

**Correspondence to**
Dr Piotr Ozieranski;
p.ozieranski@bath.ac.uk

## ABSTRACT

**Objectives** To examine the characteristics of pharmaceutical payments to healthcare and patient organisations in the four UK countries. Compare companies spending the most; types of organisations receiving payments and types of payments in the four countries. Measure the extent to which companies target payments at the same recipients in each country and whether it differs depending on the type of recipient.

**Design** Cross-sectional comparative and social network analysis.

**Setting** England, Scotland, Wales, Northern Ireland.

**Participants** 100 donors (pharmaceutical companies) reporting payments to 4229 recipients (healthcare organisations and patient organisations) in 2015.

**Main outcome measures** For each country: payment totals and distribution; average number of common recipients between companies; share of payments to organisations fulfilling different roles in the health ecosystem and payments for different activities.

**Results** Companies prioritised different types of recipient and different types of activity in each country. There were significant differences in the distribution of payments across the four countries, even for similar types of recipients. Recipients in England and Wales received smaller individual payments than in Scotland and Northern Ireland. Overall, targeting shared recipients occurred most frequently in England, but was also common in certain pockets of each country's health ecosystem. We found evidence of reporting errors in Disclosure UK.

**Conclusions** Our findings suggest a strategic approach to payments tailored to countries' policy and decision-making context, indicating there may be specific vulnerabilities to financial conflicts of interest at subnational level. Payment differences between countries may be occurring in other countries, particularly those with decentralised health systems and/or high levels of independence across its decision-making authorities. We call for a single database containing all recipient types, full location details and published with associated descriptive and network statistics.

## INTRODUCTION

Some of the major pharmaceutical companies spend more on marketing than on the development of products.[1–3] Industry marketing efforts include payments to physicians, which are seen to boost innovation and efficiency

## STRENGTHS AND LIMITATIONS OF THIS STUDY

⇒ This is the first study to compare pharmaceutical industry payments in England, Scotland, Wales and Northern Ireland.
⇒ Our analysis created a new database combining payments disclosed in Disclosure UK with individual company disclosures of payments to patient organisations.
⇒ We used social network analysis to facilitate a systematic subnational comparison of payments.
⇒ One key limitation is that the data are from 2015 so trends in payment amounts, types and targets could not be assessed.

in healthcare[4] but also generate concerns about individual financial conflicts of interest (COIs), influencing prescribing choices[5] and leading to patient harms.[6] Payments to healthcare and patient organisations have also been seen to generate institutional financial COIs around policy and programme decision making. An institution's primary goals may be unduly influenced by a secondary interest,[7] which can be more damaging than individual COIs.[7–9] COIs are defined in terms of the risk of undue influence and not actual bias or misconduct,[9] but institutional COIs have been linked to increased prescriptions of drugs with unproven safety,[8] distorting research agendas,[10] threatening the objectivity of professional education[7] and compromising independence.[11] These observations have highlighted what has been called the pharmaceutical industry's 'web of influence', in which companies 'sustain large networks to gather, create, control and disseminate information'.[12]

The potential to distort public health research and policy to favour commercial interests above patients' has led to increased policy scrutiny,[13 14] including the introduction of self-regulatory payment disclosure requirements for pharmaceutical companies in Europe.[15] Such measures are intended to aid transparency and reduce undue influence

BMJ

on clinical and policy decisions resulting from COIs. This article combines and analyses disclosure data to better understand the depth, breadth and structure of industry payments and compares them in the four countries of the UK. Comparative analysis can illustrate novel ethical and governance problems[16] or reveal that recognised problems are common across countries,[17] which our systematic examination of the extent and diversity of payments reported by pharmaceutical companies explores.

## Disclosure of industry payments

In the USA, pharmaceutical industry disclosures of payments to physicians and teaching hospitals were made mandatory in 2013,[18] and subsequent research has examined payments to physicians,[19–30] with institutional payments to hospitals largely ignored.[31] Payments to patient organisations, defined as not-for-profit institutions that primarily represent the needs of patients and/or caregivers,[32] have been seldom explored in the USA[33 34] as their disclosure is not regulated by the state or industry.

The prevalence of self-regulation in Europe is associated with very different disclosure rules to the USA.[35 36] Since 2012, the European trade association, the European Federation of Pharmaceutical Industries and Associations (EFPIA), has mandated that pharmaceutical companies publish annual disclosures of their payments to patient organisations on their websites.[37] Subsequent studies revealed extensive funding in the UK[38 39] and the Nordic countries.[40 41] However, transparency remains limited by a lack of standardised reporting requirements and limited oversight,[42] which are associated with payment under-reporting by both donors and recipients.[39]

In separate self-regulatory arrangements,[43] disclosures of payments to healthcare organisations, defined by the industry as healthcare, medical or scientific associations or organisations such as hospitals, clinics, foundations or universities,[44] have been mandated since 2015. In the UK, these are reported annually in a centralised database, Disclosure UK, hosted by the industry trade body, the Association for the British Pharmaceutical Industry (ABPI). Most research attention has been on the poor accessibility and quality of the data,[35] noting lack of standardisation of recipients[4 25] and inadequate details about individual payments' purpose.[4] These make tracking and analysing the payments complicated and time-consuming, hindering the principle aim of improving transparency.

Our study is the first, to our knowledge, to systematically combine industry data on payments to healthcare organisations and patient organisations as the self-regulatory codes allow them to be reported separately. Analysing them together enables us to better assess the breadth and depth of the 'web of influence', and gain insight into potential reinforcement effects of payments to multiple and diverse organisations that have separate yet overlapping interests, including providing patient care and support,[31 38] involvement in policy making[13 45 46] and conducting clinical research.[9 38]

## Regional differences in industry payments

Another aspect of the industry's web of influence largely unexplored in Europe is whether and how it is structured around regional differences in payments. Little is known about strategic targeting of particular fields of healthcare provision and/or decision making, nor about possible effects on COIs in regional policy making. Regionally targeted payments may have direct policy effects 'upstream', such as commissioning (the planning, prioritising and purchasing of public health services)[47] and 'downstream', such as bearing greater influence on organisational priorities and day-to-day practices.

Emerging US research has found significant differences in the distribution of payments between states,[20 48–50] including by state size[51] and political composition,[52] indicating that demographics and the organisation and regulation of healthcare matter. The first regional analysis in Europe revealed differences in the total value and type of payments prioritised in eight countries.[17] Most recently, a UK study found headquarter distance from country capitals predicts patient organisations' dependence on pharmaceutical company funding.[53] To date, research has not considered the locations of patient and healthcare organisations as the reporting requirements do not extend to disclosing country locations.[17 25 28]

However, the UK presents a crucial case for this type of analysis given its importance as a pharmaceutical market,[54] large value of payments compared with other European countries[17] and vast charitable sector comprising many potential recipients.[55] England, Scotland, Wales and Northern Ireland have four distinct health systems, with substantial autonomy to determine health policies and services.[56–58] They also differ demographically—population size is largest in England and smallest in Northern Ireland[59] and health outcomes are highest in England and lowest in Scotland.[60] The demographic and health system differences could be associated with how industry engages with different healthcare sectors.

We know that pharmaceutical companies prioritise payments to different types of healthcare organisations in the UK,[25 28] however, commercially patterned inequalities, including dominant funders or types of recipients, may be more pronounced subnationally in the in the smallest UK countries yet hidden by UK-level analysis to date.[16] Studies have started recognising the country distinction, focusing on payments to healthcare organisations in England,[47 61] but cross-country comparisons have not yet been conducted. Comparative insights could also help understand whether similar patterns are occurring in other European countries with highly decentralised healthcare setups, including Germany[62] and Spain.[63]

In this article, we apply social network analysis (SNA), which offers new insights into industry marketing tactics.[61 64 65] SNA can reveal areas of the healthcare ecosystem where connections between companies, measured by the number of payment recipients companies have in common, are most prevalent. Prevalent connections may highlight industry marketing efforts in

pockets of each of the UK's four health systems, including indicating areas of competition between companies[66–68] and revealing areas where companies are seeking to enhance their visibility.[61 69]

We integrate and analyse data from Disclosure UK and disclosures of payments to patient organisations to examine patterns in pharmaceutical company payments to organisations in the UK healthcare ecosystem. Specifically, our objectives are to:

- ► Examine the characteristics of payments to healthcare and patient organisations in the four countries.
- ► Compare the top donors financially in each country.
- ► Identify similarities and differences in the types of payments and in the types of organisations receiving payments in the four countries.
- ► Measure the extent to which companies target payments at the same recipients in each country and whether it differs depending on the type of recipient.

## METHODS
### Data sources
Our primary data sources are publicly available pharmaceutical industry transparency disclosures from 2015. Corresponding to relevant ABPI[70] and EFPIA Codes,[37] pharmaceutical companies disclose payments to healthcare organisations and to patient organisations separately.

Payments to healthcare organisations are disclosed in a centralised database, Disclosure UK, published annually by the ABPI. Payments are disclosed with recipient name, payment type (donations and grants, costs of events, joint working and consultancy—see online supplemental file 1) and value, and some address information. We use the 2015 version of Disclosure UK and focus on non-R&D payments to healthcare organisations (R&D payments are reported as one lump sum per company without named recipients).[25 43]

Payments to patient organisations are only available on individual pharmaceutical company websites and are usually presented as a PDF file and include recipient name, payment description and payment value.[42] We extracted the payments to patient organisations data into a single database, standardising names and identifying headquarter addresses as part of another project.[38] We detail our approach to data cleaning these data elsewhere.[25 38]

### Dataset preparation and integration
We followed several steps to prepare the Disclosure UK and patient organisation datasets for analysis. First, we merged the two datasets (see online supplemental file 2 for data integration flow chart). Second, as Disclosure UK provides incomplete addresses, we conducted independent web searches on each payment recipient to determine which UK country they are based in. We used the same methodology to determine patient organisations' locations. Third, we excluded payments to patient organisations duplicated in the two datasets and identified patient organisations incorrectly reported as healthcare

organisations in Disclosure UK. Fourth, we coded the patient organisation descriptions to match the codes used by Disclosure UK (online supplemental file 1).

Fifth, as part of a previous study,[25] we standardised recipient names for almost 20 000 payment entries and inductively categorised them based on their function within the healthcare system (eg, service provider) and their sector (eg, public or private) (see online supplemental file 3) for comprehensive definitions and examples of organisations). For the current study, we introduced patient organisations. Recipient types (with the most frequently occurring example) included in our analysis are:

#### Providers of health services
- ► Alternative providers of health services (eg, community interest companies providing health services).
- ► Healthcare commissioning, planning and regulatory organisations (eg, clinical commissioning groups).
- ► Private sector healthcare providers (eg, private healthcare groups).
- ► Public sector primary care providers (eg, general practitioner (GP) surgeries).
- ► Public sector secondary and tertiary care providers (eg, National Health Service (NHS) trusts).

#### Representative organisations
- ► Formal bodies representing healthcare professionals or patients (eg, local medical committees).
- ► Patient organisations (eg, multipurpose patient organisations).
- ► Professional organisations (eg, multiprofessional or multistakeholder organisations).

#### Other organisations
- ► Charities and other third-sector organisations (excludes providers of health services, professional organisations and patient organisations) (eg, charitable trusts providing educational events for healthcare professionals).
- ► Education and research providers (eg, universities).
- ► Private companies other than providers of health services (eg, providers of medical communications or training services).
- ► Public administration and providers of public services (eg, local authorities).
- ► Recipients unclear (when no information could be found).

### Analysis
We calculated the total and median value of payments in each country and recipient type. The Shapiro-Wilks test of normalcy found the data to be non-normal in each country, therefore non-parametric Kruskal-Wallis tests (adjusted for ties) were used to check for between-country differences in the distribution of payments overall and in the different recipient types. Dunn's post-hoc pairwise analyses (with Bonferroni's correction for multiple comparisons) were conducted to identify where differences were present between countries and recipient

types. Kruskal-Wallis and Dunn's tests do not assume equal sample sizes[71] and have been conducted on similar industry disclosure data.[72–74] Statistical significance was set at $p \leq 0.05$.

SNA was used to measure the number of payment recipients that were common between pairs of pharmaceutical companies (density) and across all companies (degree centrality). Density measures the overall level of connection in a network and can be used to compare the structure of different groups.[75] It produces two outputs—average value (average number of recipients each pair of companies shares)[76] and average weighted degree (average of the total number recipients each company shares with other companies). The higher these values, the more frequently a multiple companies target the same recipients in a given network.[77] For example, a density score of 1.194 tells us that all pairs of companies in the network funded an average of 1.2 recipients in common. Degree centrality, on the other hand, provides a score for each individual company based on the number of recipients in common it shares with other companies in the network—the higher the score, the more recipients a company shares.[75 78] For example, if a company has a degree centrality score of 320, they funded the same recipient as another company 320 times.

We compare the number of common recipients companies have in each country overall and when targeting different recipient types in each country. SNA requires data to be structured as a matrix, therefore, we transformed the payment data into a series of matrices of pharmaceutical companies with ties based on the number of recipients each company shared with other companies in each country and recipient type. To identify which companies targeted the same recipients, each matrix consisted only of the companies making at least one payment (regardless of whether or not they shared any recipients). We conducted separate network analyses on each of the four countries as the findings would otherwise be highly influenced by England's data as the largest network.

Data were processed in Microsoft Excel. The dataset underpinning our analysis is published in the Bath Research Data Archive.[79] We analysed the data descriptively in SPSS V.27 (IBM) and Microsoft Excel. We conducted SNA in UCINET V.6.[77] Country networks were visualised in Gephi V.0.9.2.

## Outcome measures
### Breadth of payments in each country
First, we explored the payment characteristics in each country. We measured the total and median values and the number of: payments, recipients and companies. We adjusted the total value by population size for comparison. We also compared the distribution of payments between each country using Kruskal-Wallis tests.

Second, we identified the top 10% of companies making payments in each country and compared the payment strategies of the companies paying the most in each country.

### Depth of payments in each country
Third, we assessed companies making payments to the same recipients by measuring the average number of common recipients between each pair of companies (density).

Fourth, we scrutinised which companies dominate the payment networks in each country by identifying the number of recipients that each company had in common with every other company (degree centrality).

### Structure of payments in each country
Fifth, for each country, we identified which type of recipient was prioritised. To do this, we measured and compared the proportion of payments received by each recipient type. We also compared the distribution of payments to each recipient type using Kruskal-Wallis tests to determine whether payments to similar types of recipients differ between countries.

Sixth, we examined whether companies making payments to each recipient type in each country made payments to the same organisations by measuring the average number of recipients each pair of companies share (density).

Seventh, for each country, we assessed which types of payments were prioritised through identifying the proportion of different payment types. We also compared the four types of payments using Kruskal-Wallis tests to identify differences in the distribution of payments.

### Disclosure accuracy
Finally, as a secondary outcome we measured the number of patient organisations, alongside the number and value of payments, that were incorrectly disclosed as healthcare organisations in Disclosure UK.

## Patient and public involvement
The study did not involve patients.

## RESULTS
We structure our findings consistent with the order of the outcome measures outlined above. First, we explore the breadth, depth and structure of payments in each country. While there is inevitable overlap between these framing terms, this will be signposted throughout. We also examine overall accuracy of disclosures.

### Breadth of payments in the four countries
The total value and number of payments, the number of recipients and the number of companies making payments were consistent with the size of each country, with England receiving the highest and Northern Ireland the lowest (this was maintained after adjusting for population size—see table 1).

### Between-country differences in payment values
There was a statistically significant difference in the distribution of individual payments between the four countries, $\chi^2(3) = 50.127$, $p \leq 0.001$. Dunn's post-hoc comparisons

**Table 1** Value and number of payments, number of companies and recipients, and top donors in integrated dataset

| Descriptive statistic | England | Scotland | Wales | Northern Ireland |
|---|---|---|---|---|
| Country population 2015*—n | 54 786 300 | 5 373 000 | 3 099 100 | 1 851 600 |
| Total value—£ | 52 445 615 | 3 649 749 | 1 987 703 | 518 000 |
| Total value—£ (adjusted for population size)† | 957 037 | 675 880 | 641 194 | 272 632 |
| Payments—n | 18 190 | 1370 | 990 | 311 |
| Recipients—n | 3575 | 282 | 216 | 156 |
| Companies—n | 100 | 72 | 64 | 42 |
| Median payment value (IQR)—£ | 280 (665.5) | 400 (685.3) | 300 (658.2) | 475.20 (1164.4) |
| Value of payments to healthcare organisations—£ | 40 217 772 | 3 029 365 | 1 887 918 | 474 795 |
| Value of payments to patient organisations—£ | 12 227 843 | 620 384 | 99 784 | 43 206 |

*Data obtained from the Office for National Statistics, values correct for mid-2015.
†Total value of payments divided by the population size.

showed that this difference was driven by significantly higher median payments (table 1) being made in Scotland (p≤0.001) and Northern Ireland (p≤0.001) than England. Payment size also varied significantly between Northern Ireland-Wales (p≤0.000), Scotland-Wales (p=0.001) and Northern Ireland-Scotland (p=0.004).

### Top donors in each country

The companies spending most in each country also revealed different approaches to payments (see online supplemental file 4). The top donors generally made larger payments in Wales and multiple smaller payments in Northern Ireland. Pfizer was consistently a top donor measured by value and volume of payments in all four countries, indicating an approach to payments focused on breadth. At the country level, in England, Novartis was the second biggest donor characterised by large payments; similar patterns characterised Biogen's payments in Scotland and Wales. England, Scotland and Northern Ireland all had at least one top donor not featuring as a top donor in another country, indicating some companies' payments may be more targeted regionally than others.

### Depth of engagement in the four countries

Companies making payments in England had the highest number of common recipients—an average of 6–7 recipients (table 2), implying a significant concentration of

shared interest around a spectrum of organisations. Companies, on average, had at least one recipient in common with another company in Scotland and Wales, and were least connected in Northern Ireland (table 2), indicating that in smaller countries, company interest in particular recipients is more concentrated. The average weighted degree density score shows the average number of recipients a company shares with all companies in the network, where similarly the highest score was observed in England (664.36 recipients) and lowest in Northern Ireland. The visualised networks are in online supplemental file 5.

The data also indicated variation in the depth of payments at the company level, as some companies focused collectively on particular recipients and some companies targeted a broader set of organisations with exclusive funding. Pfizer consistently targeted the same recipients as other companies most frequently in every country. Pfizer's degree centrality score of 3394 in England shows that the company funded the same organisation as another company 3394 times in the year (table 2). Many of the most connected companies (see online supplemental file 6) were similar in England, Scotland and Wales. However, Northern Ireland's top 10 most connected companies were more varied and featured smaller companies, suggesting that a cluster

**Table 2** Pharmaceutical company connections in each country measured by common recipients

| Network measure* | England | Scotland | Wales | Northern Ireland |
|---|---|---|---|---|
| Density—average value (average number of recipients in common between two companies) | 6.71 | 1.24 | 1.13 | 0.42 |
| Density—average weighted degree (average number of recipients in common for all companies in the network) | 664.36 | 88.39 | 71.06 | 17.38 |
| Company with highest degree centrality score (number of recipients a company has in common with all other companies in the network) | Pfizer (3394) | Pfizer (319) | Pfizer (206) | Pfizer (63) |

*Calculations were conducted on valued networks which means they consider the number of common recipients and not just the presence of a shared recipient. Networks include only companies making payments in each country.

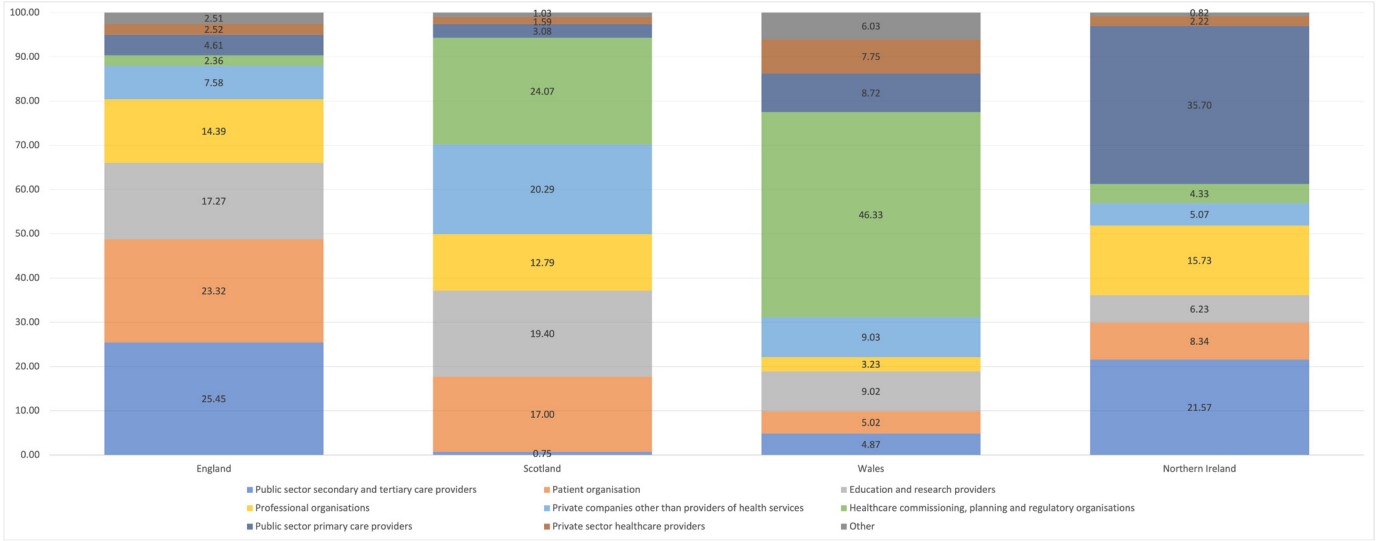

**Figure 1** Percentage of payments to recipient types per country.

of companies had a unique interest Northern Ireland's health system. Further, differences between top donors and topmost connected companies in each country highlight potentially divergent strategies in targeted funding. For example, Merck Sharpe and Dohme was highly connected in every country but was not a top donor.

Coupled, the SNA and descriptive data provides evidence that some companies prioritise breadth of payments, targeting a broader spectrum of organisations, while other companies prioritise depth, targeting recipients which seem important or 'popular' across the industry and potentially competing with other companies for visibility.

### Structure of payments in each country

#### Structural differences in targeted recipient types between countries

The share of the total value of payments received by recipient types revealed diverse funding strategies in each country (figure 1). In Wales and Scotland, industry targeted funding 'upstream' at healthcare commissioning, planning and regulatory organisations, primarily each country's local health boards that plan and deliver NHS services.[80 81] In Wales, they received just under half of all payments—£920 980.22 (46.38% of Wales' total payments, see online supplemental file 7 for values and online supplemental file 8 for top recipients). In Scotland, they received £878 333.57 (24.13%). Notably, the two Scottish health boards serving the fewest people received no payments. In England and Northern Ireland, funding was targeted 'downstream'. England's public sector secondary and tertiary care providers, namely consisting of NHS trusts which provide hospital and sometimes community healthcare services to residents,[82] received the most funding (£13 349 779.1–25.56%). In Northern Ireland, public sector primary care providers, primarily GP practices, were targeted with the most funding (£184 903.09–35.72%).

There were statistically significant differences in the distribution of payments, indicating that payment values vary between the four countries even when the recipient type is the same (see table 3). Post-hoc analyses maintained the significant differences, except for in patient organisations (see online supplemental file 9).

Patient organisations were a major target of payments, especially in England and Scotland (table 3). Professional organisations, including societies and groups of healthcare professionals, were prioritised in England, Scotland and Northern Ireland, with small yet significant differences in payment values. Consistent with the locations of the top UK universities, industry targeted education and research providers in England (median = £1000) and Scotland (£1152) where payments were also significantly higher than Wales (£336). Public sector primary care providers, primarily GP practices, received a very small proportion of the total funding in England and Scotland, yet had the most individual recipients in all four countries, suggesting smaller per-recipient payment totals. This is further reflected in the median values per recipient, which were significantly lower in England (£435) and Scotland (£435) than Wales (£800) and Northern Ireland (£600).

#### Extent of company connections in targeted recipient types in each country

Companies shared 5.8 common recipients on average among England's public sector secondary and tertiary care providers (table 4), which also received the most funding. These patterns could be a function of the number of research-active NHS trusts located in England,[83] meaning service providers might be very effective at getting donor funds, but also suggest a high degree of targeting by industry. Notably, although healthcare commissioning, planning and regulatory organisations, primarily clinical commissioning groups responsible for the planning and purchasing of local healthcare services,[84] received very

**Table 3** Differences in payment sizes between countries

| Recipient type | Median (IQR)—£ | | | | P value |
| --- | --- | --- | --- | --- | --- |
| | England | Scotland | Wales | Northern Ireland | |
| Alternative providers of health services | 200 (160–394) | 700 (360–1100) | n/a | 600 (550–600) | 0.004* |
| Charities and other third-sector organisations† | 223.52 (157–487.2) | 1700 (377.5–6250) | 120 (120–180) | n/a | 0.001* |
| Education and research providers | 1000 (333.34–4798.40) | 1152 (400–2880) | 336 (175.2–1000) | 1100 (873.75–3525) | 0.001* |
| Formal bodies representing healthcare professionals | 200 (160–259.8) | 213.6 (211.4–214.8) | 120 (96–142) | 1066.67 (933.33–1200.00) | <0.000* |
| Healthcare commissioning, planning and regulatory organisations | 208.17 (160–307.2) | 240 (131.18–500) | 225 (114.24–486.39) | 1500 (470.8–4087) | 0.008* |
| Patient organisations | 4000 (500–11 104) | 1000 (253.68–9745) | 747.93 (500–2000) | 650 (600–1450) | 0.011* |
| Private companies other than providers of health services | 300 (196.8–598.92) | 1200 (350–5528.88) | 1475 (216–6600) | 4179.38 (1152.19–7687.5) | <0.000* |
| Private sector healthcare providers | 240 (166–588) | 647.54 (206.65–1500) | 440 (360.94–1732) | 38.49 (28.9–485) | <0.000* |
| Professional organisations | 500 (240–3200) | 450 (285.67–980) | 400 (280–800) | 600 (320–1784) | 0.001* |
| Public administration and providers of public services | 394.4 (224.45–546.67) | 540 (200–600) | n/a | n/a | 0.238 |
| Public sector primary care providers | 434.5 (193.6–869) | 434.5 (202.23–651.75) | 800 (434.5–1152) | 600 (434.5–1600) | <0.000* |
| Public sector secondary and tertiary care providers | 233.17 (141.87–500) | 300 (189.75–612) | 253.66 (200 - 954) | 288 (163.4–490.13) | 0.055 |

*Statistically significant.
†Excluding providers of health services, professional organisations and patient organisations.
n/a, not available.

**Table 4**  Density scores for valued recipient type networks in each country

| Recipient type | Density scores* | | | |
|---|---|---|---|---|
| | **England** | **Scotland** | **Wales** | **Northern Ireland** |
| Alternative providers of health services | 0.339† | 0.500 | – | 0.000 |
| Charities and other third-sector organisations | 0.510 | 0.333 | 0.476 | – |
| Education and research providers | 1.194 | 0.727 | 0.675 | 1.000 |
| Formal bodies representing healthcare professionals | 1.293 | 0.000 | 0.400 | 0.000 |
| Healthcare commissioning, planning and regulatory organisations | 2.523 | 1.578 | 1.634 | 0.133 |
| Patient organisations | 0.337 | 0.200 | 0.109 | 0.209 |
| Private companies other than providers of health services | 0.312 | 0.121 | 0.071 | 0.000 |
| Private sector healthcare providers | 0.416 | 0.167 | 0.167 | 0.067 |
| Professional organisations | 0.611 | 0.244 | 0.114 | 0.038 |
| Public administration and providers of public services | 0.022 | 0.300 | 0.000 | – |
| Public sector primary care providers | 0.893 | 0.038 | 0.124 | 1.600 |
| Public sector secondary and tertiary care providers | 5.819 | 1.309 | 1.000 | 0.826 |

*Density scores measure the average number of common recipients between two companies. The network matrix for each recipient type consisted only of companies making payments. Dashes indicate no payments were made. Scores of 0.000 indicate all recipients received payments from one company only.
†Example interpretation: a score of 0.339 indicates that each company making payments to alternative providers of health services funded, on average, 0.3 recipients in common with another company.

little funding in England, companies frequently target the same recipients, indicating that low funding does not infer an absence of interest.

In Scotland and Wales, companies targeted the same healthcare commissioning, planning and regulatory organisations most frequently, consistent with the financial prioritisation. In Northern Ireland, the density score for public sector primary care providers was higher than the other countries, suggesting some companies have overlapping interests in specific recipients in pockets of Northern Ireland's primary care system. In Wales, Scotland and Northern Ireland in particular, these patterns of common recipients pose a potentially greater risk to certain areas of the healthcare ecosystem becoming vulnerable to influence given the much smaller population the organisations serve.

### Prioritised payment types in each country

Another dimension of structure that differed between countries was the type of payments (figure 2). Donations and grants, such as medical and educational goods, were consistently prioritised, however, there was notable diversity between countries among the remaining payment types. Payments for joint working, defined as initiatives involving shared investment by the NHS and pharmaceutical companies,[85] varied from 19.61% of all payments in Wales to 2.29% in Northern Ireland; fees for service and consultancy varied from 33.78% in Scotland to 4.86% in Northern Ireland; and contributions to costs of events, such as science or medical focused conferences and educational events, ranged from 31.87% in Northern Ireland to 18.58% in Wales.

There was a statistically significant difference between the distribution of payments for costs of events (p=0.000), which were lowest in Wales (£223) and highest in Northern Ireland (£478), and donations and grants (p≤0.000), which were lowest in Northern Ireland (£435) and highest in England (£960). Differences in fees for service and consultancy (p=0.995) and joint working (p=0.261) were non-significant (see online supplemental file 10).

### Accuracy of disclosures

We found evidence of pharmaceutical companies misinterpreting disclosure requirements when we integrated the Disclosure UK and patient organisation data (see online supplemental file 3 for data integration flowchart). We identified 341 payments (1.71% of all payments to organisations in Disclosure UK) to 116 patient organisations (2.88% of all organisations in Disclosure UK) worth £2 458 931.99 (5.21% of the total) incorrectly disclosed as healthcare organisations in Disclosure UK. Of these payments, 50 (14.66%) were duplicated in the patient organisation and Disclosure UK data, which were excluded to ensure no payment was counted twice.

## DISCUSSION
### Principal findings

Our findings offer insights into the pharmaceutical industry's strategic approach to payments tailored to the policy and decision-making context between, and even within, each country. Our findings also indicate that the pharmaceutical industry's 'web of influence'[14] can be

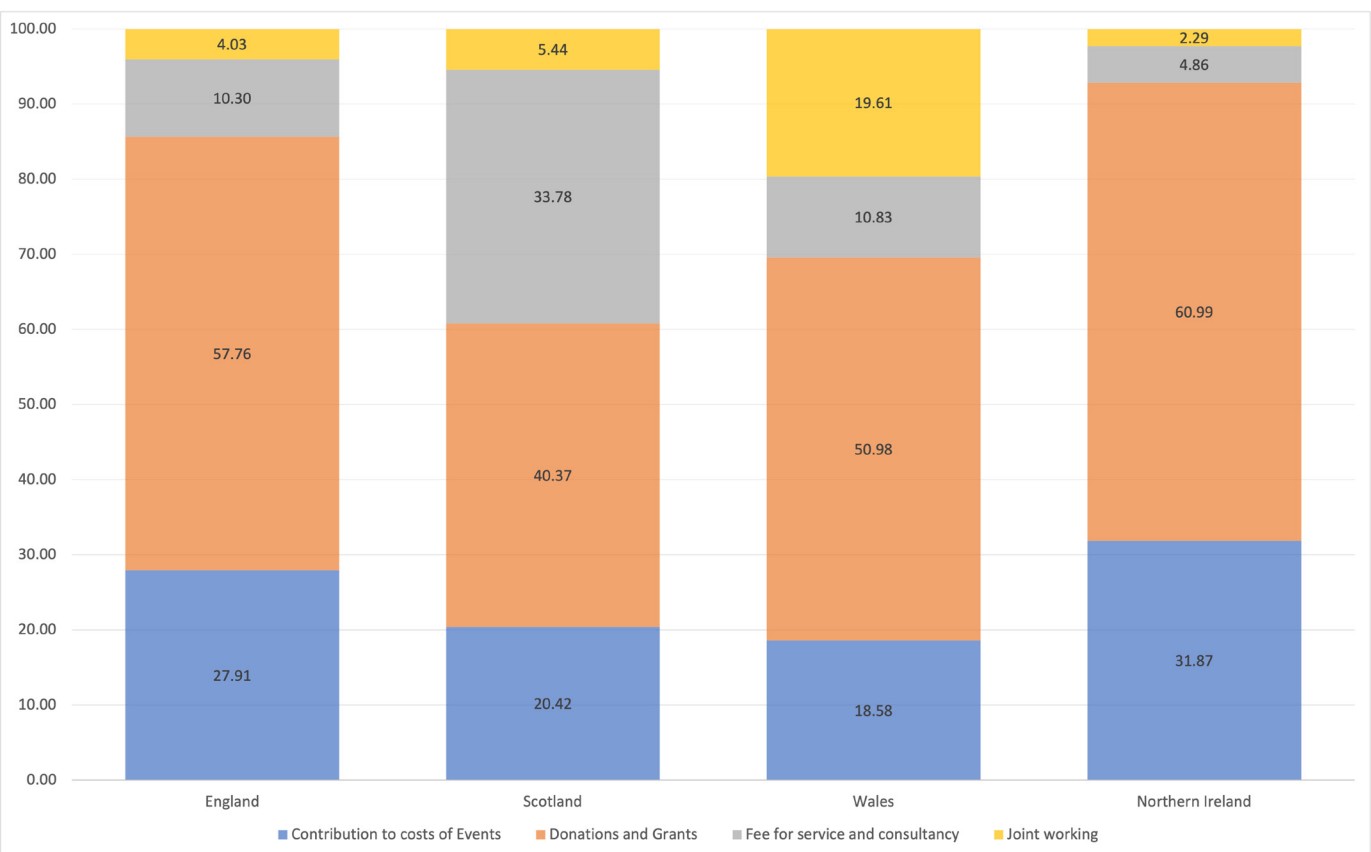

**Figure 2** Percentage of total value for each payment type.

relatively structured and aligned with key within-country differences in health system design and processes, as well as cross-nationally. Our comparative analysis illustrates novel ethical and governance problems as well as commonalities across countries and confirms concerns that UK-level analysis[25 38] obscures important regional payment variations and recipient vulnerabilities.[16] The oversight of strategic specificity is important not least because key decisions about commissioning of health services are taken within each country.[47 61]

**Findings in context**

Our findings align with previous comparative analyses of payments to teaching hospitals[31] and healthcare professionals in the USA, which show significant payment differences between regions.[20 48 49 86] Our findings also mirror those from a comparative study of industry payments to patient organisations in Denmark and Sweden, where larger payments were more frequent in the smallest country,[16] suggesting a consistent industry strategy of targeting smaller locations with larger payments.

The concentration of payments among a few companies in each country was also consistent with previous studies of patient organisations[16 38 87] and healthcare organisations.[25 31 61] We identified Pfizer as a top donor, targeting many 'popular' recipients in all four UK countries, however, it remains unclear if this relates to a particular product launch,[40 88] a new push relative to emerging competition, or reflects a consistent trend.

Further interpretation would be facilitated by longitudinal analysis. There were also differences in the companies providing the most funding, particularly in Northern Ireland where the top donors were similar to those making payments to healthcare organisations in the Republic of Ireland[28] rather than the other three UK countries, indicating that some companies may strategically target organisations on the island. One isolated case was Napp Pharmaceuticals, which featured as both a top donor and top-most connected company uniquely in Northern Ireland, suggesting that specific companies can dominate payment networks in relation to smaller countries under the radar. These instances may have direct implications for public health. For example, Napp Pharmaceuticals is an opioid manufacturer[89] and opioid manufacturers in the USA have been known to leverage targeted funding, including to teaching hospitals,[31] to increase opioid prescribing.[90]

Discrepancies in the types of payments prioritised also point towards subnational vulnerabilities in each country's healthcare ecosystem. In Wales, the prioritisation of joint working raises concerns around the extent of pharmaceutical industry involvement in healthcare design. Joint working arrangements are intended to bring benefits to patients, the NHS and companies, however, many of these projects mention increasing the use of company products,[91] potentially serving as an alternative avenue for industry marketing. Similarly, in Northern Ireland, costs

of events were higher than the other countries, pointing towards an alternative channel for industry involvement in continuing medical education in a country with fewer professional organisations or large universities. This pattern of frequent event payments was also observed in the Republic of Ireland,[28] further indicating island-specific trends.

## Lessons for transparency

The transparency concerns we identified are consistent with previous studies of pharmaceutical industry disclosure practices in the UK[4 42] and Europe.[35 41] Although the UK's self-regulatory payment disclosure system is the most robust in Europe,[17 92] our analysis confirms earlier concerns about some payments being disclosed on the incorrect platform and thereby preventing their correct identification by policymakers, regulators and members of the public.[39 42] Our findings indicate that some instances of under-reporting[39] may be explained by confusion about where to report.

These issues, coupled with the extensive additional research required to standardise and categorise recipient types and their locations in the UK, indicate that the self-regulatory system is incomplete and requires better integration. This could be achieved through a single standardised database comprising all pharmaceutical industry payments and combining the highest standards of reporting as they currently apply to, separately, healthcare and patient organisations. For example, EFPIA requires individual disclosures of payments to patient organisations to include descriptions of funded activities,[42] a provision that should be extended to healthcare organisations. As a minimum, compulsory recipient identifiers should be introduced[35] to reduce the substantial forensic work involved in cleaning these data and encourage longitudinal comparisons. Echoing calls in the USA for state-specific disclosure policies,[51] Disclosure UK and disclosures of payments to patient organisations need to be adapted to better capture the distinction between payments in England, Scotland, Wales and Northern Ireland.

While baseline improvements in data accessibility and quality are imperative, a central database should also contain associated analytics, including descriptive and network statistics. Otherwise, we run the risk that pharmaceutical companies themselves gain more from the payment disclosure system than the public, as companies use disclosures to inform and fine-tune their marketing efforts.[67 93]

## Strengths and limitations

This is the first study to jointly analyse payments to healthcare and patient organisations, which was made possible by the current UK transparency provisions. It also is the first of its kind to explore payments across the four UK countries. To date, the spotlight has been on individual COIs, which may downplay the systemic problem of a broader institutional culture whereby industry funding is embraced and industry interests can be advanced.[14 90] However, our study has limitations. We focus only on 2015 data due to the substantial time required to prepare the Disclosure UK data for effective analysis, particularly categorising recipients to make them distinguishable and identifying recipient countries. We can assume the patterns are maintained over time as the overall payment values have remained stable each year,[38 94] however, longitudinal analysis would confirm this. Also, we could not determine whether sharing recipients was accidental or intentional, nor did we measure the impact of these payments.

## CONCLUSION

Regional variability in payments has implications for subnational policy making[51] and it appears that there are specific vulnerabilities to institutional COIs arising at a subnational level. These payment differences may also be occurring in other countries, particularly those with decentralised health system structures and/or high levels of independence across their decision-making authorities. Future research could examine factors contributing to regional payment differences to better inform future government or industry policies to mitigate against undue influence.

**Contributors** ER conceived and designed the study, managed, analysed and interpreted the data, and drafted the article. PO, the guarantor for the article, conceived and designed the study, provided supervision, and drafted the article. EC provided supervision and drafted the article.

**Funding** ER's work was supported by a +3 PhD Studentship award match-funded (50%) by the Economic and Social Research Council and the University of Bath. This work forms part of the PhD project. PO's work was supported by grants from The Swedish Research Council for Health, Working Life and Welfare (FORTE), no. 2016-00875, and The Swedish Research Council (VR), no. 2020-01822.

**Competing interests** ER and EC have no conflicts of interests to declare. PO's PhD student was supported by a grant from Sigma Pharmaceuticals, a UK pharmacy wholesaler and distributor (not a pharmaceutical company). The PhD work funded by Sigma Pharmaceuticals is unrelated to the subject of this paper.

**Patient and public involvement** Patients and/or the public were not involved in the design, or conduct, or reporting, or dissemination plans of this research.

**Patient consent for publication** Not applicable.

**Provenance and peer review** Not commissioned; externally peer reviewed.

**Data availability statement** Data are available in a public, open access repository. All data relevant to the study are included in the article or uploaded as online supplementary information. The authors of this study agree to share data underpinning this study in the form of an Excel database available from the University of Bath Research Data Archive. The raw data poses no risk to anonymity of individuals as it draws on publicly available reports concerning financial transfers between organisations. The reference for this dataset is: Rickard, E., Ozieranski, P., 2023. Dataset for 'Comparing pharmaceutical company payments in the four UK countries: a cross-sectional and social network analysis'. Bath: University of Bath Research Data Archive. https://doi.org/10.15125/BATH-01239.

terminology, drug names and drug dosages), and is not responsible for any error and/or omissions arising from translation and adaptation or otherwise.

**ORCID iDs**
Emily Rickard http://orcid.org/0000-0003-2251-0127
Piotr Ozieranski http://orcid.org/0000-0002-2023-3288

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
