## [Reviewer comments · BMJ Open]

ARTICLE DETAILS

TITLE (PROVISIONAL)	Comparing pharmaceutical company payments in the four UK countries: a cross-sectional and social network analysis
AUTHORS	Rickard, Emily; Carmel, Emma; Ozieranski, Piotr

VERSION 1 – REVIEW

REVIEWER	Lexchin, Joel York University, School of Health Policy & Management In 2019-2021, Joel Lexchin received payments for writing a brief on the role of promotion in generating prescriptions for Goodmans LLP and from the Canadian Institutes of Health Research for presenting at a workshop on conflict-of-interest in clinical practice guidelines. He is a member of the Foundation Board of Health Action International and the Board of Canadian Doctors for Medicare. He receives royalties from University of Toronto Press and James Lorimer & Co. Ltd. for books he has written.
REVIEW RETURNED	18-Feb-2022

GENERAL COMMENTS	This is a very dense paper that explores an important topic, geographic variation in drug company payments within a single country. Although the authors rightly identify the limitations in their methodology the collection and analysis of data represents an enormous effort and substantially contributes to the understanding of different strategies employed by drug companies in making payments. 1. Page 3, line 28: What is the situation in Europe different from?2. Page 3, lines 29-31: Please describe what the EFPIA requirements are for reporting about payment to patient organizations and any weaknesses in the requirement.3. Do the authors have any examples of where a patient organization and a healthcare organization may have collaborated to influence policy or is this just a hypothetical concern at this point?4. Page 3, last line: The term "commissioning" should be defined for non-UK readers.5. Page 4, line 12: Something is missing in the sentence starting "Most recently a UK found..."
--

REVIEWER	Ozaki, Akihiko Minamisoma Municipal General Hospital, Department of Surgery The reviewer receives personal fees from MNES Inc.
REVIEW RETURNED	13-Mar-2022

GENERAL COMMENTS	Many thanks for giving me an opportunity to review this paper. It is an important paper working on ICOs with drug companies. I believe that some additional works would enhance an implication of this work. Major comments 1) A description of the Introduction is not sufficient with regards to the specific objectives and outcome measures employed in this study. 2) The objectives in the main text and abstracts were written vaguely. I believe that they should reflect what was evaluated in this study more accurately. 3) An interpretation of outputs social network analyses is difficult. A little more explanation would facilitate the readers' understanding. 4) The policy implication of this study should be more specific in the last part of the manuscript. Minor 1) The authors used drug companies and pharmaceutical companies to explain the same thing. This should be consistent. 2) There were some formatting errors and typos. That should be corrected in the revised manuscript.
---

REVIEWER	Grundy, Quinn University of Toronto, Lawrence S Bloomberg Faculty of Nursing
REVIEW RETURNED	13-Jun-2022

GENERAL COMMENTS	The authors conducted a cross-sectional, descriptive analysis of two data sources containing details of pharmaceutical industry payments to healthcare organizations and patient groups in 4 UK countries. This analysis is novel in that it looks at payments to organizations (instead of individuals) and undertakes a more regional analysis, comparing payment patterns across UK countries. The authors' social network analysis approach is also novel and potentially an important contribution. Please note that I do not have the expertise to evaluate the appropriateness of the statistical methods, but I do have familiarity with social network analysis. Overall, the methods appear to be appropriate and well-executed, however, the reporting is high on detail, descriptive statistics, and supplementary material, which quickly overwhelms the insights and findings that this study is positioned to generate. Currently, this analysis of transparency reports provides an interim analysis toward an understanding of industry influence within health systems. With further interpretation, a more streamlined reporting of results, an explicit rationale and justification of methods and outcomes, I believe this could be an important contribution to the literature. I had a few major comments: The data are from 2015 and justification for their relevance is needed to understand whether and how these insights are of
--

	interest/value currently and also whether they are transferrable to other settings. The authors make the argument for a regional analysis and the appropriateness of the UK for this undertaking. I think this argument could be further strengthened by bringing it full circle at the end of the introduction – what can be learned from a regional analysis of UK countries and what are the potential impacts for the UK or beyond? If this can also be better reflected in the abstract, I think this would strengthen the impact of the paper. Much of the contextual detail that suggests why the comparative analysis is useful comes only in the Discussion. I would suggest incorporating more of the contextual details pertaining to the 4 countries when these differences are presented in the results section and explaining to the reader what it means. Similarly, the ‘so what’ is not well-reflected in the study objective. I suggest that the authors articulate the overarching goal that reflects the study’s importance and then specific aims that get at the descriptive analysis. Because these data are now a few years old, it is very important to articulate the implications/value of detecting these patterns and what they mean rather than just the number crunching. While the social network analysis is a novel approach to transparency reports, the authors do not well explain in the introduction or methods the value of this methodology or what knowledge can be generated from this approach. Though they have included a supplementary file, I think some detail about the assumptions used to make the analytic decisions is needed in the manuscript, tailored for a generalist audience. My biggest concern is that the authors only conducted a social network level at the regional level – so essentially 4 different networks – and we do not have the benefit of understand connections across regions. I also question the choice to exclude companies with no ties from the network analysis and wonder if a two-stage approach (the entire network and then analysis of a connected core) might give a more accurate picture? The introduction relies on the concept of “institutional COIs” but this is never explicitly defined. I concur with Marks (who you cite) that often conflict of interest is used as shorthand for industry influence/interference more broadly. The issues of influence/independence and accountability are less well articulated. Key definitions are missing from the body of the manuscript including the key recipient groups, payment types etc. Though the supplementary files contain ample detail, the reader needs the authors to distill this information into the key concepts and categories necessary to understand what the authors did and what it means. For example, I would have appreciated a clear definition of “healthcare organization” and “patient organization” as you will use them in the paper. In the methods, I often felt like there was too much information and yet still, the information I wanted was in the supplementary files. I wonder if a series of concise tables could communicate the key
--	---

	information in the methods section without resorting to lengthy supplementary files. The results too are high on level of detail but the meaning or interpretation becomes a bit lost. I would suggest that the authors incorporate more of their own interpretation as topic sentences and then support this with the evidence they have collected. Some of this interpretation is currently in the Discussion, but is more appropriately results. The discussion instead, should place these results in the context of the literature and suggest implications for practice/policy. For example, what does the descriptive evidence about duplicate reporting mean? Was this a high level of duplication? A common occurrence? Or relatively accurate? Similarly, the upstream/downstream differences in funding patterns in the Discussion section should accompany these results so the reader understands what is presented, including its context. The Discussion then provides a great deal of contextual information and it is not always clear whether this is grounded in the data. For example, the discussion of primary care privatization in Northern Ireland. In many ways, 'less is more' and the authors should consider which of the policy implications they wish to prioritize for greater impact. The authors might consider organizing the material by level of policymaking or policy body/process they aim to target, or by theme (e.g. transparency, managing industry relations, etc) The conclusion/policy implications adds further detail without added clarity and I suggest this conclusion should be a simple paragraph with closing thoughts. Minor comments: Abstract:  - Upon first reading, the 'main outcome measure' "overlap of payments in Disclosure UK. . ." was not sufficiently self-explanatory for me; the other main outcome measures were much clearer - Similarly, the first sentence of the Results required more contextual knowledge about the transparency datasets for full understanding. Could this be rephrased as a more general critique of transparency (e.g. "We found numerous errors in . . .") - While the abstract Results reported descriptive statistics, it is unclear what they mean, especially for an international audience – could the meaning/import be instead reported here as the high-level synopsis? Strengths and limitation:  • For an international audience, see if you can instead communicate why this is listed as a strength. What can be learned?: "The first study to comparatively assess pharmaceutical industry payments across the UK countries" Introduction  - When you mention the EIPFIA requirements, could you specify whether the mandatory reporting requirements stipulated where/how these payments were reported? - Is a word missing here: "Most recently a UK found headquarter distance from country capitals to be a significant predictor of patient organisations' dependence on drug companies funding." Also, what did they find (in terms of direction of association)? - The introduction contains many long and complex sentences – it would help the readability to break some of these up into shorter sentences and perhaps to include some information in a table (e.g. demographics of different UK countries).
--	--

	Methods - I did not understand the distinctions fully between the patient organizations, charities and third-sector organizations in the section on Data Integration – can you define these terms? Results - I did not understand this sentence: “mirroring observations of frequent individual COIs held by professional organisations”. What is an individual organizational COI? Thank you for the opportunity to review this manuscript, which I believe is an important contribution.
--	---

REVIEWER	van de Vijver, David Erasmus MC, Viroscience
REVIEW RETURNED	11-Nov-2022

GENERAL COMMENTS	Rickard and Ozieranski have analyzed publicly available data on drug company payments in four UK countries. The authors found that 100 drug companies paid 4,299 organizations almost 60 million GBP. Overall payments were relatively small with median values ranging between 280 GBP in England 475.2 GBP in Northern Ireland. 1. I am concerned about the use of the Kruskal Wallis test given the unequal sample size between the four UK countries. Most payments were made to organizations in England (18190 out of a total of 20861 or 87.2% of all payments). In addition, Northern Ireland receives the highest median amount and includes the largest variation in the amount paid (as reflected with the widest IQR of all four countries, see Table 1). Kruskal Wallis compares the mean rank for each country. Consequently, a relatively high or low amount paid outside of England may therefore have a larger impact on the statistical outcome as the high (or low) rank is divided by a much smaller number as compared to England. I would therefore advise to repeat the analysis and restrict the English data to a randomly chosen number of about 10-20% of all payments. 2. The data was restricted to 2015. Are more recent data available? 3. Unfortunately, I could not read the bar charts and kindly request if the authors could increase the font sizes. 4. Please add the median amount paid to recipient categories by country in Table 3. Without these amounts it is rather difficult to assess which countries are driving the differences.
--

VERSION 1 – AUTHOR RESPONSE

Reviewer: 1
Dr. Joel Lexchin, York University

Reviewer 1's comments	Authors' response
1. Page 3, line 28: What is the situation in Europe different from?	We have clarified that the situation in Europe is different from the US (due to self-regulation in the former). The revised sentence now reads: “The prevalence of self-regulation in Europe is associated with very different disclosure rules to the US^{35 36.}” (page 3, line 36)

2. Page 3, lines 29-31: Please describe what the EFPIA requirements are for reporting about payment to patient organizations and any weaknesses in the requirement.	We have added some additional detail to this section (specifically we now specify that the disclosure requirements include recipient names and payment values, and that payments must be disclosed on individual companies' websites). The revised sentence reads: “Since 2012, the European trade association, the European Federation of Pharmaceutical Industries and Associations (EFPIA), has mandated that pharmaceutical companies publish annual disclosures of their payments to patient organisations on their websites³⁷.” (page 3, lines 37-40) We have also introduced additional information about the weaknesses of pharmaceutical industry disclosures of payments to patient organisations. The revised sentence reads: “However, transparency remains limited by a lack of standardised reporting requirements and limited oversight which are associated with payment under-reporting by both donors and recipients” (page 3, lines 41-43)
3. Do the authors have any examples of where a patient organization and a healthcare organization may have collaborated to influence policy or is this just a hypothetical concern at this point?	Our data does not provide evidence on collaboration between organizations that receive pharmaceutical company payments. We have clarified that patient organisations and healthcare organisations have many separate yet overlapping (replaced the word ‘shared’ for ‘overlapping’ to avoid the impression that we are talking about collaboration) interests and both form part of the industry’s ‘web of influence’: “Analysing them together enables us to better assess the breadth and depth of the ‘web of influence’, and gain insight into potential reinforcement effects of payments to multiple and diverse organisations that have separate yet overlapping interests including providing patient care and support^{31 38}, involvement in policy-making^{13 45 46}, and conducting clinical research^{9 38} (page 4, lines 10-14)
4. Page 3, last line: The term "commissioning" should be defined for non-UK readers.	We have added a definition of commissioning when the term is first mentioned in the Introduction. The sentence reads: “Regionally targeted payments may have direct policy effects ‘upstream’, such as commissioning (the planning, prioritising, and purchasing of public health services); and ‘downstream’, such as bearing greater influence on organisational priorities and day-to-day practices.” (page 4, lines 20-23)
5. Page 4, line 12: Something is missing in the sentence starting "Most recently a UK found..."	We have added the missing word ‘study’ to complete this sentence. (page 4, line 29)

Reviewer 2, Dr. Akihiko Ozaki, Minamisoma Municipal General Hospital

We would like to express our appreciation for the time you took to review our manuscript. We appreciate the insightful comments you have provided and have enjoyed engaging with them.

Reviewer 2's comment	Authors' response
Major comments	
1) A description of the Introduction is not sufficient with regards to the specific objectives and outcome measures employed in this study.	We have now made extensive changes to the Introduction to provide more contextual information preceding the study objectives and outcome measures. Specifically, we have added the following additional paragraphs to describe the importance of comparative analysis and the regional aspect of the objectives: “Comparative analysis can illustrate novel ethical and governance problems¹⁶ or reveal that recognised problems are common across countries¹⁷, which our systematic examination of the extent and diversity of payments reported by pharmaceutical companies explores” (page 3, lines 23-26) “Little is known about strategic targeting of particular fields of healthcare provision and/or decision-making, nor about possible effects on potential COIs in regional policy-making. Regionally targeted payments may have direct policy effects ‘upstream’, such as commissioning (the planning, prioritising, and purchasing of public health services)⁴⁷; and ‘downstream’, such as bearing greater influence on organisational priorities and day-to-day practices.” (page 4, lines 18-23) “...however commercially patterned inequalities, including dominant funders or types of recipients, may be more pronounced sub-nationally in the in the smallest UK countries yet hidden by UK-level analysis to date¹⁶.” (page 4, lines 44-46) We have also added additional information justifying the importance of looking at connections between companies via common recipients (i.e. the SNA). The updated text is: “SNA can reveal areas of the healthcare ecosystem where connections between companies, measured by the number of payment recipients companies have in common, are most prevalent. Prevalent connections may highlight industry marketing efforts in pockets of each of the UK’s health systems, including indicating areas of competition between companies^{66 67 68} and revealing areas where companies are seeking to enhance their visibility^{61 69}.” (page 5, lines 7-12)

2) The objectives in the main text and abstracts were written vaguely. I believe that they should reflect what was evaluated in this study more accurately.	We now phrase our objectives much more precisely and explicitly to avoid any confusion about what our study intended to look at. These are now expressed as bullet points as follows: We have revised both the abstract and the introduction with the same, clearer objectives. In the Abstract these read: “Objectives To examine the characteristics of pharmaceutical payments to healthcare and patient organisations in the four UK countries. Compare companies spending the most; types of organisations; and types of payments in the four countries. Measure the extent to which companies target payments at the same recipients in each country and whether it differs depending on the type of recipient.” (page 1, lines 25-29) The revised section in the Introduction now presents the Objectives in bullet points and specifies to set them out more clearly: “We integrate and analyse data from Disclosure UK and disclosures of payments to patient organisations to examine patterns in pharmaceutical company payments to organisations in the UK healthcare ecosystem. Specifically, our objectives are to:  • examine the characteristics of payments to healthcare and patient organisations in the four countries • compare the top donors financially in each country • identify similarities and differences in the types of payments and in the types of organisations receiving payments in the four countries • measure the extent to which companies target payments at the same recipients in each country and whether it differs depending on the type of recipient” (page 5, lines 15-25)
3) An interpretation of outputs social network analyses is difficult. A little more explanation would facilitate the readers' understanding.	We have addressed the comment in a number of ways throughout the manuscript. We have incorporated the information that was only available in a supplementary file into the methods (and removed the supplementary file) to clarify what the social network analysis calculates. We have also added a hypothetical example and interpretation of density and degree centrality in the methods. The heavily revised section now reads: “SNA was used to measure the number payment recipients that were common between pairs of pharmaceutical companies (density) and across all companies (degree centrality). Density measures the overall level of connection in a network and can be used to compare the structure of different groups⁷⁵. It produces two outputs – average value (average number of recipients each pair of companies shares⁷⁶) and average weighted degree (average of the total number recipients each company shares with other companies). The higher these values, the more frequently a multiple companies target the same recipients in a given network⁷⁷. For example, a density score of 1.194 tells us that all pairs of companies in the network

funded an average of 1.2 recipients in common. Degree centrality, on the other hand, provides a score for each individual company based on the number of recipients in common it shares with other companies in the network – the higher the score, the more recipients a company shares^{75 78}. For example, if a company has a degree centrality score of 320, they funded the same recipient as another company 320 times.” (page 7, lines 9-21)

Secondly, in the results section we now include interpretation of the findings as they are presented to give them meaning. We spell out what the SNA statistics tell us and clarify what the numbers indicate, for example:

“The data also indicates variation in the depth of payments at the company level, as some companies focus collectively on particular recipients and some companies target a broader set of organisations with exclusive funding. Pfizer consistently targeted the same recipients as other companies most frequently in every country. Pfizer’s degree centrality score of 3,394 in England shows that the company funded the same organisation as another company 3,394 times in the year (Table 2).” (page 10, lines 21-26)

Another example:

“Companies shared 5.8 common recipients on average among England’s public sector secondary and tertiary care providers (Table 4), which also received the most funding. These patterns could be a function of the number of research-active NHS trusts located in England⁸², meaning service providers might be very effective at getting donor funds, but also suggest a high degree of targeting by industry. Notably, although healthcare commissioning, planning and regulatory organisations, primarily clinical commissioning groups responsible for the planning and purchasing of local health care services⁸³, received very little funding in England, companies frequently target the same recipients, indicating that low funding does not infer an absence of interest.” (page 12, lines 5-13)

We also regularly remind the reader what the numbers mean, for example:

“In Northern Ireland, the density score for public sector primary care providers was higher than the other countries, suggesting some companies have overlapping interests in specific recipients in pockets of Northern Ireland’s primary care system.” (page 12, lines 15-19)

In Table 2 we have introduced more detail and adjusted the phrasing (replaced ‘ties’ with ‘recipients in common’) to ensure these terms are used consistently throughout the manuscript to avoid confusion. The detail we have added is in bold below:

*“Density – average value (average number of **recipients in common** between two companies)*

	Density – average weighted degree (average number of recipients in common for all companies in the network) Company with highest degree centrality score (number of recipients a company has in common with all other companies in the network)” (page 10, Table 2) Under Table 4 we have introduced another note offering an example interpretation of one of the values: ***Example interpretation: a score of 0.339 indicates that each company making payments to alternative providers of health services funded, on average, 0.3 recipients in common with another company”. (page 12, Table 4)
Minor comments	
1) The authors used drug companies and pharmaceutical companies to explain the same thing. This should be consistent.	Thank you for this observation. We have made sure pharmaceutical companies are referred to consistently throughout the manuscript and supplementary files.
2) There were some formatting errors and typos. That should be corrected in the revised manuscript.	Thank you for pointing this out. We have proofread the manuscript.

Reviewer: 3 Dr. Quinn Grundy, University of Toronto

We would like to thank you for your extremely thoughtful and helpful comments on our manuscript. We appreciate you taking the time to review our work and we have endeavoured to address and engage with all points raised.

Reviewer 3's comment	Authors' response
Major comments	
The data are from 2015 and justification for their relevance is needed to understand whether and how these insights are of interest/value currently and also whether they are transferrable to	Thank you for raising this point. The reason that the data is from 2015 is due to the scale of what is involved in preparing this data for analysis. Therefore, our analysis represents a starting point and, although future analysis can use our methodology for data cleaning and preparation, it is still likely to be a time-consuming process given the volume (in excess of 20,000 rows) of payments that need to be standardised and coded. This is why we stress in the conclusion that a

other settings.	single database with recipient identifiers, payment descriptions, and specific location details are required. We have added a comment in the introduction as to how time-consuming preparing this type of data for analysis is: “These make tracking and analysing the payments complicated and time-consuming, hindering the principle aim of improving transparency.” (page 4, lines 5-6) In the introduction we have added a sentence about how our comparative analysis could apply to other countries: “Comparative insights could also help understand whether similar patterns are occurring in other European countries with highly decentralised healthcare set-ups, including Germany⁶² and Spain⁶³.” (page 5, lines 2-4) In the methods we have added that preparing the data for analysis involved inductively categorising almost 20,000 rows of data. “Fifth, as part of a previous study²⁵ we standardised recipient names for almost 20,000 payment entries and inductively categorised them based on their function within the healthcare system (e.g. service provider) and their sector (e.g. public or private) (see Supplementary File 3 for comprehensive definitions and examples of organisations).” (page 6, lines 12-15) Our work is the first of its kind and future work can compare with newer data to assess patterns. As such, we have mentioned in the discussion that longitudinal analysis will be important to assess trends in the data over time: “Further interpretation would be facilitated by longitudinal analysis” (page 14, line 35-36) As well as here: “We can assume the patterns are maintained over time as the overall payment values have remained stable each year^{38 98}, however longitudinal analysis would confirm this.” (page 16, lines 3-5) And finally, when presenting our suggestions for how disclosure transparency could be improved (as lack of transparency is currently the biggest barrier to longitudinal analysis in the UK):
--

	“As a minimum, compulsory recipient identifiers should be introduced³⁵ to reduce the substantial forensic work involved in preparing these payments for analysis and encourage longitudinal comparisons.” (page 15, lines 29-31)
The authors make the argument for a regional analysis and the appropriateness of the UK for this undertaking. I think this argument could be further strengthened by bringing it full circle at the end of the introduction – what can be learned from a regional analysis of UK countries and what are the potential impacts for the UK or beyond? If this can also be better reflected in the abstract, I think this would strengthen the impact of the paper. Much of the contextual detail that suggests why the comparative analysis is useful comes only in the Discussion. I would suggest incorporating more of the contextual details pertaining to the 4 countries when these differences are presented in the results section and explaining to the reader what it means.	This comment has helped us to strengthen the manuscript hugely. We have incorporated much of the contextual detail from the discussion into the introduction and results sections to explain to, firstly, strengthen the rationale for regional analysis and, secondly, to explain to the reader what the results mean as and when they are presented. In the introduction, we have added the following sentences to justify regional analysis in the UK. These additions also pre-empt the results and discussion sections to further add to the overall ‘story’ of the manuscript. “Comparative analysis can illustrate novel ethical and governance problems¹⁶ or reveal that recognised problems are common across countries¹⁷, which our systematic examination of the extent and diversity of reported payments by pharmaceutical companies explores.” (page 3, lines 23-26) “Little is known about strategic targeting of particular fields of healthcare provision and/or decision-making, nor about possible effects on potential COIs in regional policy-making. Regionally targeted payments may have direct policy effects ‘upstream’, such as commissioning (the planning, prioritising, and purchasing of public health services); and ‘downstream’, such as bearing greater influence on organisational priorities and day-to-day practices.” (page 4, lines 18-23) “...however commercially patterned inequalities, including dominant funders or types of recipients, may be more pronounced sub-nationally in the in the smallest UK countries yet hidden by UK-level analysis to date¹⁶.” (page 4, lines 44-46) As mentioned in response to your earlier comment, we have also added a sentence with the international relevance of our research as it relates to countries with similar health system set-ups: “Comparative insights could also help understand whether similar patterns are occurring in other European countries with highly decentralised healthcare set-ups, including Germany⁶² and Spain⁶³.” (page 5, lines 2-4) This has also been incorporated into the abstract: “Payment differences between countries may be occurring in other countries, particularly those with decentralised health systems and/or high levels of independence across its decision-making authorities.” (page 2, lines 10-12)

In the conclusion of the abstract we have also introduced further interpretation of the findings to show the importance and relevance of the study:

“Our findings suggest a strategic approach to payments tailored to countries’ policy and decision-making context, indicating there may be specific vulnerabilities to potential financial conflicts of interest at sub-national level.” (page 2, lines 6-8)

We have also incorporated text from the discussion into the results section, particularly to give meaning and context to the types of recipient receiving payments. For example:

Having introduced the ‘upstream/downstream’ concept in the introduction, we refer to it again in the results to explain the findings. Here we refer to ‘upstream’ targeting and provide detail about the type and role of targeted organisations:

“In Wales and Scotland, industry targeted funding ‘upstream’ at healthcare commissioning, planning and regulatory organisations, primarily each country’s local health boards that plan and deliver NHS services^{78 79}.... Notably, the two Scottish health boards serving the fewest people received no payments.” (page 11, lines 6-9 and 12-13)

Here we discuss the ‘downstream’ targeting in England and Northern Ireland and provide detail about the type and role of organisations:

“In England and Northern Ireland, funding was targeted ‘downstream’. England’s public sector secondary and tertiary care providers, namely consisting of NHS trusts which provide hospital and sometimes community healthcare services to residents⁸⁰, received the most funding (£13,349,779.1 – 25.56%). In Northern Ireland, public sector primary care providers, primarily general practitioner practices, were targeted with the most funding (£184,903.09 – 35.72%).” (page 11, lines 13-18)

We provide more detail about what professional organisations are and also link the findings with some context about universities in the UK:

“Professional organisations, including societies and groups of healthcare professionals, were prioritised in England, Scotland and Northern Ireland, with significant but negligible differences in payment values. Consistent with the locations of the top UK universities, industry targeted education and research providers in England (median = £1000) and Scotland, (£1,152) where payments were also significantly higher than Wales (£336).” (page 11, lines 28-32)

	We offer interpretation of very high shared recipients among certain recipients in England: “Companies shared 5.8 common recipients on average among England’s public sector secondary and tertiary care providers (Table 4), which also received the most funding. These patterns could be a function of the number of research-active NHS trusts located in England⁸³, meaning service providers might be very effective at getting donor funds, but also suggest a high degree of targeting by industry. Notably, although healthcare commissioning, planning and regulatory organisations, primarily clinical commissioning groups responsible for the planning and purchasing of local health care services⁸⁴, received very little funding in England, companies frequently target the same recipients, indicating that low funding does not infer an absence of interest.” (page 12, lines 5-13) In the results we also point towards the implications of the findings for specific areas of each country’s health system: “In Wales, Scotland and Northern Ireland in particular, these patterns of common recipients pose a potentially greater risk to certain areas of the healthcare ecosystem becoming vulnerable to influence given the much smaller population the organisations serve.” (page 12, lines 19-22) Finally, in the results we have now incorporated information about the payment types that was previously not presented until the discussion: “Donations and grants, such as medical and educational goods, were consistently prioritised, however there was notable diversity between countries among the remaining payment types. Payments for joint working, defined as initiatives involving shared investment by the NHS and pharmaceutical companies⁸², varied from 19.61% of all payments in Wales to 2.29% in Northern Ireland; fees for service and consultancy varied from 33.78% in Scotland to 4.86% in Northern Ireland; and contributions to costs of events, such as science or medical focused conferences and educational events, ranged from 31.87% in Northern Ireland to 18.58% in Wales.” (page 13, lines 11-19)
Similarly, the ‘so what’ is not well-reflected in the study objective. I suggest that the authors articulate the overarching goal that reflects the study’s importance and then specific aims that get at the descriptive analysis. Because these data are now a few years old, it is very important to articulate the implications/value of detecting these patterns and what they mean rather than just	To better reflect the ‘so what’ of the study we highlight the importance and relevance of conducting comparative analysis on payments. We emphasise the potential insights that can be gained from comparative analysis. Firstly, early on in the introduction: “Comparative analysis can illustrate novel ethical and governance problems¹⁶ or reveal that recognised problems are common across countries¹⁷, which our systematic examination of the extent and diversity of reported payments by pharmaceutical companies explores.” (page 3, lines 23-26) And then followed up on later in the introduction:

the number crunching.	"Little is known about strategic targeting of particular fields of healthcare provision and/or decision-making, nor about possible effects on potential COIs in regional policy-making. Regionally targeted payments may have direct policy effects 'upstream', such as commissioning (the planning, prioritising, and purchasing of public health services); and 'downstream', such as bearing greater influence on organisational priorities and day-to-day practices." (page 4, lines 18-23) Further justification of the so what is provided now when we suggest that dominant funders or types of recipients in smaller nations may otherwise be hidden: "We know that pharmaceutical companies prioritise payments to different types of healthcare organisations in the UK^{25 28}, however commercially patterned inequalities, including dominant funders or types of recipients, may be more pronounced sub-nationally in the in the smallest UK countries yet hidden by UK-level analysis to date¹⁶." (page 4, lines 43-47) We have also rephrased the latter paragraphs in the introduction to better reflect what we are looking at. We have added some additional references on previous applications of SNA with similar data to show that this is an emerging and important method to show areas of concentrated engagement and also areas where industry may be seeking to be more visible or even the occurrence of competition amongst companies. This new paragraph reads: "In this article, we apply social network analysis (SNA) which offers new insights into industry marketing tactics^{64 65 61}. SNA can reveal areas of the healthcare ecosystem where connections between companies, measured by the number of payment recipients companies have in common, are most prevalent. Prevalent connections may highlight industry marketing efforts in pockets of each of the UK's health systems, including indicating areas of competition between companies^{66 67 68} and revealing areas where companies are seeking to enhance their visibility^{61 69}." (page 5, lines 6-12) We also now present out objectives in a much more concise and clear way to further clarify what our research is looking at: "We integrate and analyse data from Disclosure UK and disclosures of payments to patient organisations to examine patterns in pharmaceutical company payments to organisations in the UK healthcare ecosystem. Specifically, our objectives are to:  • examine the characteristics of payments to healthcare and patient organisations in the four countries • compare the top donors financially in each country
--

	 • identify similarities and differences in the types of payments and in the types of organisations receiving payments in the four countries • measure the extent to which companies target payments at the same recipients in each country and whether it differs depending on the type of recipient” (page 5, lines 14-24) The new summary of our findings under ‘Principal findings’ now also offers further detail into what our study provides and its importance: “Our findings offer insights into the pharmaceutical industry’s strategic approach to payments tailored to the policy and decision-making context between, and even within, each country. Our findings also indicate that the pharmaceutical industry’s ‘web of influence’¹⁴ can be relatively structured and aligned with key within-country differences in health system design and processes, as well as cross-nationally. Our comparative analysis illustrates novel ethical and governance problems¹⁶ as well as commonalities across countries¹⁷ and confirms concerns that previous analysis of industry payments at the UK level^{25 38} obscured important regional payment variations and recipient vulnerabilities¹⁶. The oversight of strategic specificity is important not least because key decisions about commissioning of health services are taken within each country^{47 61}.” (page 14, lines 12-21)
While the social network analysis is a novel approach to transparency reports, the authors do not well explain in the introduction or methods the value of this methodology or what knowledge can be generated from this approach. Though they have included a supplementary file, I think some detail about the assumptions used to make the analytic decisions is needed in the manuscript, tailored for a generalist audience. My biggest concern is that the authors only conducted a social network level at the regional level – so essentially 4 different networks – and we do not have the benefit of understand connections across regions.	In addressing the previous comment we have partially addressed this one. We have added additional information about what knowledge can be gained from SNA in the Introduction, specifically the potential impact it may have on industry marketing tactics through indicating areas of potential competition between companies and enhancing company visibility. Additionally, so that readers do not have to consult the supplementary files to find out more about SNA, we have removed Supplementary File 3 and incorporated the relevant text into the Methods section, whilst aiming to phrase it in a way that suits a generalist audience. We have also added hypothetical examples and interpretations of SNA values. The revised section now reads: “SNA was used to measure the number payment recipients that were common between pairs of pharmaceutical companies (density) and across all pharmaceutical companies (degree centrality). Density measures the overall level of connection in a network and produces two outputs – average value (average number of recipients each pair of companies shares⁷⁴) and average weighted degree (average of the total number recipients each company shares with other companies). The higher these values, the more frequently a multiple companies target the same recipients in a given network⁷⁵. For example, a density score of 1.194 tells us that all pairs of companies in the network funded an average of 1.2 recipients in common. Degree centrality, on the other hand, provides a score for each individual company based on the number of recipients in common it shares with other companies in the network – the higher the score, the more recipients a company shares^{76 77}. For example, if a company has a degree centrality score of 320, they funded the same recipient as another company 320 times.” (page 7, lines 9-21)

	Additionally, we further emphasise the new insights provided by SNA in the results by comparing the SNA data with the descriptive data: “Coupled, the SNA and descriptive data provides evidence that some companies prioritise breadth of payments, targeting a broader spectrum of organisations, while other companies prioritise depth, targeting recipients which seem important or ‘popular’ across the industry and potentially competing with other companies for visibility.” (page 10, lines 34-35, cont page 11, lines 1-2) In response to your final point and only conducting networks at the regional level, we have added a sentence in the Methods to explain why we conducted separate networks for each of the countries. The sentence reads: “We conducted separate network analyses on each of the four countries as the findings would otherwise be highly influenced by England’s data as the largest network.” (page 7, lines 29-31) We have also added that density is a useful tool for comparing networks (to re-emphasise the focus of the paper on the comparative angle: “Density measures the overall level of connection in a network and can be used to compare the structure of different groups⁷⁵” (page 7, lines 11-12) As the four networks relate to four separate health systems, studying connections would not be possible unless the focus was only on companies. But our objectives with regards to the SNA were to, firstly, compare connections between companies in each of the four countries and secondly, compare connections between companies targeting particular recipient types in each country. The focus of our paper is on the comparative aspect of how companies approach payments depending on the location.
I also question the choice to exclude companies with no ties from the network analysis and wonder if a two-stage approach (the entire network and then analysis of a connected core) might give a more accurate picture?	Thank you for raising this. We have now clarified that we did not exclude companies with no ties (because this is important in the context of our objectives). We excluded companies making no payments in a given category of recipients or country because we know they would not share any recipients (because they did not even make a single payment). We set out to examine the extent to which companies making payments were targeting the same recipients, and we believe that our approach provides the most accurate overall picture of this. We feel we have made this much clearer in the revised sentence and additional information in the methods as follows: “To identify which companies targeted the same recipients, each matrix consisted only of the companies making at least one payment (regardless of whether or not they shared any recipients).” (page 7, lines 27-29)
The introduction relies on the concept of “institutional COIs”	We have now added a short definition of institutional conflicts of interest in the Introduction where the term is first used. It reads:

but this is never explicitly defined. I concur with Marks (who you cite) that often conflict of interest is used as shorthand for industry influence/interference more broadly. The issues of influence/independence and accountability are less well articulated.	“Payments to healthcare and patient organisations have also been seen to generate potential institutional financial COIs around policy and programme decision-making. An institution’s primary goals may be unduly influenced by a secondary interest⁷, which can be more damaging than individual COIs⁷⁻⁹.” (page 3, lines 6-9) “Industry marketing efforts include payments to physicians, which are seen to boost innovation and efficiency in healthcare⁴ but also generate concerns about potential individual financial conflicts of interest (COIs), influencing prescribing choices⁵ and leading to patient harms⁶. Payments to healthcare and patient organisations have also been seen to generate potential institutional financial COIs around policy and programme decision-making.” (page 3, lines 3-8) We have added a sentence in the introduction to clarify that the presence of a COI does not equal influence, but rather it poses a risk: “COIs are defined in terms of the risk of undue influence and not actual bias or misconduct.” (page 3, lines 9-10) We have also added compromised independence as an implication of ICOIs: “...institutional COIs have been linked to increased prescriptions of drugs with unproven safety⁸, distorting research agendas¹⁰, threatening the objectivity of professional education⁷, and compromising independence¹¹.” (page 3, lines 10-13) In relation to influence, we have added that transparency measures have been introduced to reduce potential COIs and undue influence: “Such measures are intended to aid transparency, reducing potential conflicts of interest and undue influence on clinical and policy decisions.” (page 3, lines 20-21) And also point towards the type of influence that can occur in particular recipient types: “Regionally targeted payments may have direct policy effects ‘upstream’, such as commissioning (the planning, prioritising, and purchasing of public health services)⁴⁷; and ‘downstream’, such as bearing greater influence on organisational priorities and day-to-day practices.” (page 4, lines 20-23)
Key definitions are missing from the body of the manuscript including the key recipient groups, payment types etc. Though the supplementary files contain ample detail, the reader	We have added a more detailed definition of patient organisation in the introduction when the term is first used, and we have reiterated the definition in the methods when discussing data integration. The definition in the introduction reads:

needs the authors to distill this information into the key concepts and categories necessary to understand what the authors did and what it means. For example, I would have appreciated a clear definition of “healthcare organization” and “patient organization” as you will use them in the paper.	“Payments to patient organisations, defined as not-for-profit institutions that primarily represent the needs of patients and/or caregivers³², have been seldom explored in the US^{33 34} as their disclosure is not regulated by the state or industry” (page 3, lines 32-34) We have also added a more comprehensive definition of healthcare organisation in the introduction when it is first mentioned: “In separate self-regulatory arrangements⁴³, disclosures of payments to healthcare organisations, defined by the industry as healthcare, medical or scientific associations or organisations such as hospitals, clinics, foundations or universities⁴⁴, have been mandated since 2015.” (page 3, lines 45-47) We have renamed ‘recipient category’ to ‘recipient type’/‘type of recipient’ with the intention of removing unnecessary effort for the reader. In the methods section, we have now inserted a list of the recipient types with an example of an organisation included in each type. We think this bullet point list will help distil the information and reduce the burden on the reader. Additional detail about each recipient type is still available in the supplementary files, but this way it is not essential to access these unless the reader wants more detail. (page 6, lines 19-43) We have now made sure that the recipient types are clearly defined when first mentioned, for example when first mentioning healthcare commissioning, planning and regulatory organisations in Wales and Scotland: “In Wales and Scotland, industry targeted funding ‘upstream’ at healthcare commissioning, planning and regulatory organisations, primarily each country’s local health boards that plan and deliver NHS services^{80 81}” (page 11, lines 7-9) And secondary and tertiary care providers in England: “England’s public sector secondary and tertiary care providers, namely consisting of NHS trusts which provide hospital and sometimes community healthcare services to residents⁸², received the most funding (£13,349,779.1 – 25.56%).” (page 11, lines 14-16) Primary care providers in Northern Ireland: “In Northern Ireland, public sector primary care providers, primarily general practitioner practices, were targeted with the most funding (£184,903.09 – 35.72%).” (page 11, lines 16-18)
---	---

	Professional organisations: “Professional organisations, including societies and groups of healthcare professionals, were prioritised in England, Scotland and Northern Ireland, with significant but negligible differences in payment values.” (page 11, lines 28-30) We now include a brief description of the payments types in the results section where they are first presented rather than in the discussion. The relevant text now reads: “Donations and grants, such as medical and educational goods, were consistently prioritised, however there was notable diversity between countries among the remaining payment types. Payments for joint working, defined as initiatives involving shared investment by the NHS and pharmaceutical companies⁸⁵, varied from 19.61% of all payments in Wales to 2.29% in Northern Ireland; fees for service and consultancy varied from 33.78% in Scotland to 4.86% in Northern Ireland; and contributions to costs of events, such as science or medical focused conferences and educational events, ranged from 31.87% in Northern Ireland to 18.58% in Wales.” (page 13, lines 12-19)
In the methods, I often felt like there was too much information and yet still, the information I wanted was in the supplementary files. I wonder if a series of concise tables could communicate the key information in the methods section without resorting to lengthy supplementary files.	We have now eliminated the supplementary file discussing the SNA measures as we agree that this is important information that must be clearly described up front. As we already had four tables and 2 figures in the manuscript, we decided to include the necessary methods details in the text of the Methods section. The section now reads: “SNA was used to measure the number payment recipients that were common between pairs of pharmaceutical companies (density) and across all companies (degree centrality). Density measures the overall level of connection in a network and can be used to compare the structure of different groups⁷⁵. It produces two outputs – average value (average number of recipients each pair of companies shares⁷⁶) and average weighted degree (average of the total number recipients each company shares with other companies). The higher these values, the more frequently a multiple companies target the same recipients in a given network⁷⁷. For example, a density score of 1.194 tells us that all pairs of companies in the network funded an average of 1.2 recipients in common. Degree centrality, on the other hand, provides a score for each individual company based on the number of recipients in common it shares with other companies in the network – the higher the score, the more recipients a company shares^{75 78}. For example, if a company has a degree centrality score of 320, they funded the same recipient as another company 320 times. We compare the number of common recipients companies have in each country overall and when targeting different recipient types in each country. SNA requires data to be structured as a matrix, therefore we transformed the payment data into a series of matrices of pharmaceutical companies with ties based on the number of recipients each company shared with other companies in each country and recipient type. To identify which companies targeted the same recipients, each

	matrix consisted only of the companies making at least one payment (regardless of whether or not they shared any recipients). We conducted separate network analyses on each of the four countries as the findings would otherwise be highly influenced by England’s data as the largest network.” (page 7, lines 9-31) We have also added additional information on interpreting SNA statistics in and below the tables where they appear as well as in text when the statistics are discussed. In Table 2 we have introduced more detail and adjusted the phrasing (replaced ‘ties’ with ‘recipients in common’) to ensure these terms are used consistently throughout the manuscript to avoid confusion. The detail we have added is in bold below: Density – average value (average number of recipients in common between two companies) Density – average weighted degree (average number of recipients in common for all companies in the network) Company with highest degree centrality score (number of recipients a company has in common with all other companies in the network) (page 10, Table 2) Under Table 4 we have introduced another note offering an example interpretation of one of the values: “**Example interpretation: a score of 0.339 indicates that each company making payments to alternative providers of health services funded, on average, 0.3 recipients in common with another company” (page 13, Table 4) As mentioned above, we have also introduced a list of recipient types into the methods section so that the reader does not have to consult the supplementary files to understand what organisations we are looking at.
The results too are high on level of detail but the meaning or interpretation becomes a bit lost. I would suggest that the authors incorporate more of their own interpretation as topic sentences and then support this with the evidence they have collected. Some of this interpretation is currently in the Discussion, but is more appropriately results. The discussion instead, should place these results in the	We have changed the subheading of the section on duplicate reporting to ‘Accuracy of disclosures’ for increased clarity. We have also reworded this paragraph to more clearly express what we are describing. “Accuracy of disclosures We found evidence of pharmaceutical companies misinterpreting disclosure requirements when we integrated the Disclosure UK and patient organisation data (see Supplementary File 3 for data integration flowchart). We identified 341 payments (1.71% of all payments to organisations in Disclosure UK) to 116 patient organisations (2.88% of all organisations in Disclosure UK) worth £2,458,931.99 (5.21% of the total) incorrectly disclosed as healthcare organisations in Disclosure

context of the literature and suggest implications for practice/policy. For example, what does the descriptive evidence about duplicate reporting mean? Was this a high level of duplication? A common occurrence? Or relatively accurate? Similarly, the upstream/downstream differences in funding patterns in the Discussion section should accompany these results so the reader understands what is presented, including its context	UK. Of these payments, 50 (14.66%) were duplicated in the patient organisation and Disclosure UK data, which were excluded to ensure no payment was counted twice. ” (page 8, lines 39-46) We have now moved much of the discussion/context points into the findings alongside definitions of recipients to further explain what the results are showing. For example, the newly subtitled “Structural differences in targeted recipient types between countries” section of the results provides interpretation to aid understanding of the findings – it now reads: “The share of the total value of payments received by recipient types revealed diverse funding strategies in each country (Figure 1). In Wales and Scotland, industry targeted funding ‘upstream’ at healthcare commissioning, planning and regulatory organisations, primarily each country’s local health boards that plan and deliver NHS services^{80 81}. In Wales, they received just under half of all payments - £920,980.22 (46.38% of Wales’ total payments, see Supplementary File 7 for values and Supplementary File 8 for top recipients). In Scotland, they received £878,333.57 (24.13%). Notably, the two Scottish health boards serving the fewest people received no payments. In England and Northern Ireland, funding was targeted ‘downstream’. England’s public sector secondary and tertiary care providers, namely consisting of NHS trusts which provide hospital and sometimes community healthcare services to residents⁸², received the most funding (£13,349,779.1 – 25.56%). In Northern Ireland, public sector primary care providers, primarily general practitioner practices, were targeted with the most funding (£184,903.09 – 35.72%). ” (page 11, lines 6-18) We also brought context from the discussion into the results in the shared interest in recipients section: “In Wales, Scotland and Northern Ireland in particular, these patterns of common recipients pose a potentially greater risk to certain areas of the healthcare ecosystem becoming vulnerable to influence given the much smaller population the organisations serve. ” (page 12, lines 19-22)
The Discussion then provides a great deal of contextual information and it is not always clear whether this is grounded in the data. For example, the discussion of primary care privatization in Northern Ireland. In many ways, ‘less is more’ and the authors should consider which of the policy implications they wish to prioritize for greater impact. The authors might consider organizing the material by level of policymaking or policy body/process they aim to target, or by theme (e.g.	We have now incorporated many of the contextual points made in the discussion into the results section to give the findings more meaning and reduce repetition. We have also removed some sentences that, on reflection following your comments, we realise did not add value to the overall message of the paper and agree that less is more. We believe that our Results section now provides substantial context to the findings when they are first presented, and the Discussion section now tries to provide a generalist overview of the findings and their potential implications in practice, as well as drawing comparisons internationally and picking up on a couple of interesting company specific cases (Pfizer and Napp) as areas that warrant further longitudinal investigation.

transparency, managing industry relations, etc) The conclusion/policy implications adds further detail without added clarity and I suggest this conclusion should be a simple paragraph with closing thoughts.	For example: “The finding that the total value of payments was concentrated to a few companies in each country was also consistent with previous studies of patient organisations^{16 38 87} and healthcare organisations^{25 31 61}. We identified Pfizer as a top donor, targeting many ‘popular’ recipients in all four countries, however it remains unclear if this relates to a particular product launch^{40 88}, a new push relative to emerging competition, or reflects a consistent trend. Further interpretation would be facilitated by longitudinal analysis. There were also differences in the companies providing most funding, particularly in Northern Ireland where the top donors were similar to those making payments to healthcare organisations in the Republic of Ireland²⁸ rather than the other three UK countries, indicating that some companies may strategically target organisations on the island. One isolated case was Napp Pharmaceuticals, which featured as both a top donor and top-most connected company uniquely in Northern Ireland, suggesting that specific companies can dominate payment networks in relation to smaller countries under the radar. These instances may have direct implications for public health. For example, Napp Pharmaceuticals is an opioid manufacturer⁸⁹ and opioid manufacturers in the United States have been known to leverage targeted funding, including to teaching hospitals³¹, to increase opioid prescribing⁹⁰. (page 14, lines 33-46, cont. page 15, lines 1-2) We have also restructured the Discussion using sub-headings to provide further clarity and direct the reader. We start with principle findings, which provides a generalist overview summarising our findings and their overall implications (page 14, lines 12-21) Next we have findings in context, which link our findings to existing literature and expand upon certain cases (particularly companies as mentioned above in relation to Pfizer and Napp), some of the findings in relation to each country’s health system, and similarities between our study and other studies’ transparency findings– (page 14, lines 24-46, cont page 15, lines 1-11) We then have a lessons for transparency section which focuses on transparency compared to previous studies and our suggestions for improvement (page 15, lines 14-40) Finally, following the strengths and limitations section, we now have a conclusion which is much more concise than it was previously. It summarises the findings briefly, indicates the international relevance of our work, and points towards where future research could go (page 16, lines 10-16)
---	---

Minor comments	
Abstract	
Abstract: - Upon first reading, the ‘main outcome measure’ “overlap of payments in Disclosure UK. . .” was not sufficiently self-explanatory for me; the other main outcome measures were much clearer	We have rephrased this to more accurately explain what we looked at: “Number of payments incorrectly reported in Disclosure UK and payments reported twice in two places –Disclosure UK and disclosures on company websites” (page 1, lines 38-39)
- Similarly, the first sentence of the Results required more contextual knowledge about the transparency datasets for full understanding. Could this be rephrased as a more general critique of transparency (e.g. “We found numerous errors in . . .”)	We have amended this sentence so that it is now more general, it reads: “We found evidence of reporting errors in Disclosure UK.” (page 1, line 44)
- While the abstract Results reported descriptive statistics, it is unclear what they mean, especially for an international audience – could the meaning/import be instead reported here as the high-level synopsis?	We have shortened the results section of the abstract so it now focuses on providing a general overview and touches on the importance of the findings. It reads: “Results We found evidence of reporting errors in Disclosure UK. Companies prioritised different types of recipient and payments for different types of activity in each country. There were significant differences in the distribution of payments across the four countries, even for similar types of recipients. Recipients in England and Wales received smaller individual payments than in Scotland and Northern Ireland. Overall, targeting shared recipients occurred most frequently in England, but was also common in certain pockets of each country’s health ecosystem.” (page 1, lines 44-46, cont. page 2, lines 1-4)
Strengths and limitation:	
• For an international audience, see if you can instead communicate why this is listed as a strength. What can be learned?: “The first study to comparatively assess pharmaceutical industry payments across the UK countries”	Following a comment from the editor specifically about how the strengths and limitations section (following the abstract) was phrased, we have altered the bullet points so that they relate specifically to the methods. The revised bullet points now read:  • Ours is the first study to compare pharmaceutical industry payments in England, Scotland, Wales and Northern Ireland • Our analysis was based on a combination of payments disclosed in the Disclosure UK dataset and industry disclosures of payments to patient organisations • We use social network analysis to facilitate a systematic sub-national comparison of payments • One key limitation is that the data is from 2015

	Page 2, lines 15-23
Introduction	
When you mention the EIPFIA requirements, could you specify whether the mandatory reporting requirements stipulated where/how these payments were reported?	We have added some additional detail to this section (specifically we now specify that the disclosure requirements include recipient names and payment values, and that payments must be disclosed on individual companies' websites). The revised sentence reads: "Since 2012, the European trade association, the European Federation of Pharmaceutical Industries and Associations (EFPIA), has mandated that pharmaceutical companies publish annual disclosures of their payments to patient organisations on their websites³⁷." (page 3, lines 37-40) Similarly, we have added further clarification to the sentence where we first introduce the disclosure requirements for healthcare organisations. The revised sentence now reads: "In separate self-regulatory arrangements⁴³, disclosures of payments to healthcare organisations, defined by the industry as healthcare, medical or scientific associations or organisations such as hospitals, clinics, foundations or universities⁴⁴, have been mandated since 2015. In the UK, these are reported annually in a centralised database, Disclosure UK, hosted by the industry trade body, the Association for the British Pharmaceutical Industry (ABPI)." (page 3, lines 45-47, cont, page 4, lines 1-3)
- Is a word missing here: "Most recently a UK found headquarter distance from country capitals to be a significant predictor of patient organisations' dependence on drug companies funding." Also, what did they find (in terms of direction of association)?	The missing word 'study' has now been added. (page 4, line 29)
- The introduction contains many long and complex sentences – it would help the readability to break some of these up into shorter sentences and perhaps to include some information in a table (e.g. demographics of different UK countries).	Thank you for highlighting this. We have now substantially reworked the introduction in addressing other insightful comments about better rationalising our work and the context of the four UK countries, in doing so, we have reduced the number of long and complex sentences. As the manuscript already has four tables and two figures, we decided instead to streamline how the introduction and contextual details about each country are presented. We removed a couple of the demographic details as they confused the section with no obvious value added.
Methods	

- I did not understand the distinctions fully between the patient organizations, charities and third-sector organizations in the section on Data Integration – can you define these terms?	We have reworded and added more information about how these terms are defined in the newly named Dataset preparation and integration section. Instead of readers having to consult the supplementary files for a list of the types of recipients included we have now introduced a list, alongside an example of the most occurring specific organisations, of each type of recipient in our study. We hope this makes reading the remainder of the paper simpler and clearer. The charities and third-sector organisations example now indicates how this category differs from the patient organisation category – “e.g. charitable trusts providing educational events for healthcare professionals”. List of organisational types is on page 6, lines 19-43
Results	
- I did not understand this sentence: “mirroring observations of frequent individual COIs held by professional organisations”. What is an individual organizational COI?	We have now removed this sentence in line with the restructuring of the results and discussion sections.

Reviewer: 4
Dr. David van de Vijver, Erasmus MC

Reviewer 4’s comments	Authors’ response
Major comments	
1. I am concerned about the use of the Kruskal Wallis test given the unequal sample size between the four UK countries. Most payments were made to organizations in England (18190 out of a total of 20861 or 87.2% of all payments). In addition, Northern Ireland receives the highest median amount and includes the largest variation in the amount paid (as reflected with the widest IQR of all four countries, see Table 1). Kruskal	We have sought statistical advice from a number of resources on this important issue. The consensus was that reducing the sample size in the large group would lose power for no obvious gain anywhere else. Additionally, randomly removing payment data would mean that two people using the same test on the same data would come to different results (depending on the random payments selected). We cannot locate any literature to suggest randomly choosing a percentage of the sample when conducting a Kruskal Wallis test. To this end, we have added justification for the use of the Kruskal-Wallis test by including references that specify that unequal sample sizes are not a requirement. We have also added references to studies in the same field (analysis of

Wallis compares the mean rank for each country. Consequently, a relatively high or low amount paid outside of England may therefore have a larger impact on the statistical outcome as the high (or low) rank is divided by a much smaller number as compared to England. I would therefore advise to repeat the analysis and restrict the English data to a randomly chosen number of about 10-20% of all payments.	pharmaceutical payment disclosures in the US) which used Kruskal-Wallis tests to compare similar, highly varied, groups of payments: “Kruskal-Wallis and Dunn’s tests do not assume equal sample sizes⁷¹ and have been conducted on similar industry disclosure data^{72 73 74}.” (page 7, lines 5-6) When reflecting on your comment, we also trialled Mood’s median test in SPSS – the results were very similar to the Kruskal Wallis (i.e the significant differences were maintained), suggesting that using the mean rank does not affect the results in the large group. We hope this response shows that we have given this comment due care and consideration.
2. The data was restricted to 2015. Are more recent data available?	We appreciate you raising this point. The reason that the data is from 2015 is due to the scale of what is involved in preparing this data for analysis. Therefore, our analysis represents a starting point and, although future analysis can use our methodology for data cleaning and preparation, it is still likely to be a time-consuming process given the volume (in excess of 20,000 rows) of payments that need to be standardised and coded. This is why we stress in the conclusion that a single database with recipient identifiers, payment descriptions, and specific location details are required. We have added a comment in the introduction as to how time-consuming preparing this type of data for analysis is: “These make tracking and analysing the payments complicated and time-consuming, hindering the principle aim of improving transparency.” (page 4, lines 5-6) In the introduction we have added a sentence about how our comparative analysis could apply to other countries: “Comparative insights could also help understand whether similar patterns are occurring in other European countries with highly decentralised healthcare set-ups, including Germany⁶² and Spain⁶³.” (page 5, lines 2-4) In the methods we have added that preparing the data for analysis involved inductively categorising almost 20,000 rows of data. “Fifth, as part of a previous study³³ we standardised recipient names for almost 20,000 payment entries and inductively categorised them based on their function

	within the healthcare system (e.g. service provider) and their sector (e.g. public or private)” (page 6, lines 12-17) Our work is the first of its kind and future work can compare with newer data to assess patterns. As such, we have mentioned in the discussion that longitudinal analysis will be important to assess trends in the data over time: “We identified Pfizer as a top donor, targeting many ‘popular’ recipients in all four countries, however it remains unclear if this relates to a particular product launch⁴⁰⁸⁸, a new push relative to emerging competition, or reflects a consistent trend. Further interpretation would be facilitated by longitudinal analysis.” (page 14, lines 33-36) As well as here: “We can assume the patterns are maintained over time as the overall payment values have remained stable each year³⁸⁹⁴, however longitudinal analysis would confirm this.” (page 16, lines 3-5) And finally, when presenting our suggestions for how disclosure transparency could be improved (as lack of transparency is currently the biggest barrier to longitudinal analysis in the UK): “As a minimum, compulsory recipient identifiers should be introduced³⁵ to reduce the substantial forensic work involved in preparing these payments for analysis and encourage longitudinal comparisons.” (page 15, lines 29-31)
3. Unfortunately, I could not read the bar charts and kindly request if the authors could increase the font sizes.	Thank you for noticing this. We have now increased the font size to 12 on both of the bar charts.
4. Please add the median amount paid to recipient categories by country in Table 3. Without these amounts it is rather difficult to assess which countries are driving the differences.	We have now changed Table 3 so that it provides more useful and meaningful information to support the text in the results. We have added the median amounts (and IQRs) for each recipient type in each country, alongside the p value. We still signpost the reader to the supplementary files for the results of the post-hoc analysis which reveals which specific medians were significantly different between countries. (pages 11-12, Table 3)

VERSION 2 – REVIEW

REVIEWER	Ozaki, Akihiko Minamisoma Municipal General Hospital, Department of Surgery I receive personal fees from MNES Inc., and Kyowa Kirin outside the submitted work.
REVIEW RETURNED	17-Jan-2023

GENERAL COMMENTS	I have no further comments. Thank you for your revision.
---

REVIEWER	Grundy, Quinn University of Toronto, Lawrence S Bloomberg Faculty of Nursing
REVIEW RETURNED	19-Jan-2023

GENERAL COMMENTS	Thank you for the opportunity to re-review this manuscript, which is greatly strengthened. I have one outstanding comment, which pertains to how the objectives are reported. In the abstract and introduction, one of the main outcome measures has to do with discrepancies in reports and reporting consistency quality ("disclosure accuracy"). This does not reflect the stated revised objectives, which now well communicate the study importance. I would suggest moving all references to reporting discrepancies to a place as a secondary outcome (including in the abstract and results). I offer the following minor comments:  - The sentence in the Objective section of the abstract could be further clarified (e.g. what is meant by types of organizations - meaning types that receive payments or types of recipient organizations?): "Compare companies spending the most; types of organisations; and types of payments in the four countries" - because conflicts of interest represent a risk, I suggest removing the adjective "potential" where used in reference to conflicts of interest. There is a potential for negative outcomes, but the conflicts of interest exist objectively
--

REVIEWER	van de Vijver, David Erasmus MC, Viroscience
REVIEW RETURNED	31-Jan-2023

GENERAL COMMENTS	The authors have addressed my concerns. I remain concerned that the data is relatively old (from 2015), but the authors have included a discussion about the date of collection of their data.
--

VERSION 2 – AUTHOR RESPONSE

Reviewer: 3

Dr. Quinn Grundy, University of Toronto

Comments to the Author:

Thank you for the opportunity to re-review this manuscript, which is greatly strengthened. I have one outstanding comment, which pertains to how the objectives are reported. In the abstract and introduction, one of the main outcome measures has to do with discrepancies in reports and reporting consistency quality ("disclosure accuracy"). This does not reflect the stated revised objectives, which now well communicate the study importance. I would suggest moving all references to reporting discrepancies to a place as a secondary outcome (including in the abstract and results).

We fully agree that the disclosure accuracy is in fact a secondary outcome. We now reflect this throughout the manuscript. The following sections have been revised:

Abstract

- We have now removed the sentence about incorrect reporting from the 'Main outcome measures' in the Abstract as this is now a secondary outcome.
- We have now moved "We found evidence of reporting errors in Disclosure UK" so that it the final sentence of the Results in the Abstract rather than the first sentence (p. 2, line 2).

Methods

- In the Outcome Measures section we have moved Disclosure Accuracy to the end of the section so it is the final outcome measure. We also clarified that this was a secondary outcome measure: "Finally, as a secondary outcome we measured the number of patient organisations, alongside the number and value of payments, that were incorrectly disclosed as healthcare organisations in Disclosure UK." (p. 8, lines 24-26)

Results

- We have amended the opening paragraph of the results to reflect the changed order of outcome measures.
- We have moved the section reporting on disclosure accuracy to the end of the results section in line with the revised order of the outcome measures (p. 13, lines 29-34)

I offer the following minor comments:

- The sentence in the Objective section of the abstract could be further clarified (e.g. what is meant by types of organizations - meaning types that receive payments or types of recipient organizations?): "Compare companies spending the most; types of organisations; and types of payments in the four countries"

Thank you for pointing this out. We have now revised this sentence to read “Compare companies spending the most; types of organisations receiving payments; and types of payments in the four countries.” (p 1, lines 26-27)

- because conflicts of interest represent a risk, I suggest removing the adjective "potential" where used in reference to conflicts of interest. There is a potential for negative outcomes, but the conflicts of interest exist objectively

We appreciate you raising this important point and have now removed the ‘potential’ when describing conflicts of interest.

VERSION 3 – REVIEW

REVIEWER	Grundy, Quinn University of Toronto, Lawrence S Bloomberg Faculty of Nursing
REVIEW RETURNED	09-Mar-2023
GENERAL COMMENTS	Thank you for the opportunity to review this revised manuscript. The authors have addressed all of my previous comments and I have no further comments.

VERSION 3 – AUTHOR RESPONSE